# Crystalline silica-exposed human lung epithelial cells presented enhanced anchorage-independent growth with upregulated expression of BRD4 and EZH2 in autocrine and paracrine manners

**Motoo Katabami, Ichiro Kinoshita\*, Shin Ariga, Yasushi Shimizu, Hirotoshi Dosaka-Akita**

Department of Medical Oncology, Faculty of Medicine and Graduate School of Medicine, Hokkaido University, Sapporo, Japan

\* kinoshii@med.hokudai.ac.jp

**Data Availability Statement:** All relevant data are within the paper.

## Abstract

Crystalline silica-induced inflammation possibly facilitates carcinogenesis. Here, we investigated its effect on lung epithelium damage. We prepared conditioned media of immortalized human bronchial epithelial cell lines (hereinafter bronchial cell lines) NL20, BEAS-2B, and 16HBE14o- pre-exposed to crystalline silica (autocrine crystalline silica conditioned medium), a phorbol myristate acetate-differentiated THP-1 macrophage line, and VA13 fibroblast line pre-exposed to crystalline silica (paracrine crystalline silica conditioned medium). As cigarette smoking imposes a combined effect on crystalline silica-induced carcinogenesis, a conditioned medium was also prepared using the tobacco carcinogen benzo[a]pyrene diol epoxide. Crystalline silica-exposed and growth-suppressed bronchial cell lines exhibited enhanced anchorage-independent growth in autocrine crystalline silica and benzo[a]pyrene diol epoxide conditioned medium compared with that in unexposed control conditioned medium. Crystalline silica-exposed nonadherent bronchial cell lines in autocrine crystalline silica and benzo[a]pyrene diol epoxide conditioned medium showed increased expression of cyclin A2, cdc2, and c-Myc, and of epigenetic regulators and enhancers, BRD4 and EZH2. Paracrine crystalline silica and benzo[a]pyrene diol epoxide conditioned medium also accelerated the growth of crystalline silica-exposed nonadherent bronchial cell lines. Culture supernatants of nonadherent NL20 and BEAS-2B in crystalline silica and benzo[a]pyrene diol epoxide conditioned medium had higher EGF concentrations, whereas those of nonadherent 16HBE14o- had higher TNF-α levels. Recombinant human EGF and TNF-α promoted anchorage-independent growth in all lines. Treatment with EGF and TNF-α neutralizing antibodies inhibited cell growth in crystalline silica conditioned medium. Recombinant human TNF-α induced BRD4 and EZH2 expression in nonadherent 16HBE14o-. The expression of γH2AX occasionally increased despite PARP1 upregulation in crystalline silica-exposed nonadherent lines with crystalline silica and benzo[a]pyrene diol epoxide conditioned medium. Collectively, crystalline silica- and benzo[a]pyrene diol

**Funding:** The author(s) received no specific funding for this work.

**Competing interests:** The authors have declared that no competing interests exist.

epoxide-induced inflammatory microenvironments comprising upregulated EGF or TNF-α expression may promote crystalline silica-damaged nonadherent bronchial cell proliferation and oncogenic protein expression despite occasional γH2AX upregulation. Thus, carcinogenesis may be cooperatively aggravated by crystalline silica-induced inflammation and genotoxicity.

## Introduction

Although a recent meta-epidemiological study confirmed a positive exposure–response relationship between the definite carcinogen crystalline silica (CS) and lung cancer [1], the mechanism by which CS induces inflammation in carcinogenesis remains underexplored. Lung cancer incidence is higher in individuals exposed to CS who suffer from silicosis than in those not afflicted by this disease [2]. CS-induced lung cancer in rats is preceded by pulmonary inflammation and fibrosis [3]. Clinical data have shown that pneumoconiosis-related lung cancer develops preferentially during diffuse interstitial fibrosis-type pneumoconiosis [4]. These results support the hypothesis that silicosis leads to CS-induced lung cancer development. However, this notion was questioned by a recent meta-epidemiological study that indicated a robust exposure–response relationship between CS and the occurrence of lung cancer in individuals exposed to CS without silicosis [1]. Some *in vitro* studies have demonstrated the transformation of human bronchial cell lines after chronic exposure to CS without the aid of a CS-induced inflammatory microenvironment, such as in treatment with CS-exposed conditioned medium (CM) or CS-induced inflammatory mediators [5, 6]. Direct CS exposure causes apoptosis and DNA damage in bronchial cells [7, 8]. Recent studies have indicated that CS induces mutable DNA double-strand breaks in bronchial cells [9]. Therefore, the mechanism underlying the transformation induced by CS in these *in vitro* studies could be attributed to CS-induced DNA damage, which eventually causes genetic aberrations. In addition, there are only a few *in vivo* studies on the induction of lung cancer via CS in the absence of inflammation, making it difficult to extrapolate *in vitro* data to *in vivo* studies. Furthermore, the contribution of CS-induced inflammation during the early stages of CS-induced carcinogenesis remains unclear. To resolve the discrepancy between *in vitro* and *in vivo* studies, an *in vitro* model must show that bronchial cells, after direct exposure to CS, undergo rapid proliferation in the CS-induced inflammatory microenvironment. For this purpose, we investigated whether bronchial cell lines directly pre-exposed to CS still show enhanced growth after incubation with CM obtained from bronchial cell lines (autocrine CS CM), a phorbol myristate acetate (PMA)-differentiated THP-1 macrophage cell line, and VA13 fibroblast cell line (paracrine CS CM). As cigarette smoking has been shown to confer at least a super-additive combined effect on CS-induced carcinogenesis [10], benzo[a]pyrene diol epoxide (BPDE) CM was also investigated.

It has been demonstrated that the CS-induced inflammatory microenvironment is accompanied by the upregulation of soluble mediators such as EGF-like molecules, TNF-α, TGF-β1, and IL-6 [11–14]. However, the role of these mediators in CS-induced carcinogenesis remains unclear. Therefore, we also conducted a study focusing on the soluble factors in CS CM- and BPDE CM-incubated bronchial cell lines.

## Materials and methods

### Reagents

CS (Min-U-Sil5, particle range: 0.6–5 μm) purchased from Pennsylvania Glass Sand Corporation (Pittsburgh, PA, USA) was sterilized in its solid form by heating at 180˚C for 2 h in a dry

oven before use [15]. BPDE was purchased from the NCI Chemical Carcinogen Reference Standards Repository (Kansas City, MO, USA) [16]. Recombinant human EGF, TNF-α, TGF-β1, and IL-6 were purchased from R&D Systems (R&D Systems Inc., Minneapolis, MN, USA). Mouse anti-human EGF neutralizing antibody (NtAb), goat anti-human TNF-α NtAb, and mouse anti-human IL-6 NtAb were obtained from R&D Systems.

## Human bronchial epithelial cell lines

The SV40-immortalized human bronchial epithelial cell lines NL20 and BEAS-2B were purchased from the American Type Culture Collection (ATCC, Manassas, VA, USA). The human bronchial epithelial cell line 16HBE14o- (16HBE) immortalized with SV40 was gifted by Prof. Dieter C. Gruenert (University of California, San Francisco, CA, USA) [17]. NL20 cells were maintained in F-12K medium (GIBCO, Grand Island, NY) with 2.7 g/L glucose, 0.1 mM non-essential amino acids, 5 μg/mL insulin, 10 ng/mL EGF, 1 μg/mL transferrin, 500 ng/mL hydrocortisone, and 4% fetal bovine serum (FBS), and were passaged twice a week [18]. BEAS-2B cells, with a doubling time of approximately 26 h, were maintained in a serum-free LHC-9 medium containing 50 ng/mL EGF, and were passaged twice a week [19]. The 16HBE cells were cultured in Eagle's minimum essential medium (EMEM) (Invitrogen Life Technologies Inc., Carlsbad, CA, USA) supplemented with 10% FBS and were passaged weekly. BEAS-2B and 16HBE cells were attached and grown in flasks coated with a solution of fibronectin (1 mg/mL; Alfa Aesar, Ward Hill, MA, USA), collagen (Vitrogen 100, 2.9 mg/mL; Collagen Corp., Palo Alto, CA, USA), and bovine serum albumin (BSA Fraction V 1 mg/mL; Thermo Fisher Scientific, Waltham, MA, USA).

## Preparation of conditioned medium

Anchorage-independent growth is considered a phenotypic hallmark of neoplastic transformation [20]. Poly(hydroxyethyl) methacrylate [poly-(HEMA)]-coated plates or dishes were used to render adherent cells nonadherent, allowing the evaluation of anchorage-independent growth and the conduct of subsequent experiments. Human bronchial cell lines were plated on T-75 tissue culture plates in each medium. When the cells reached sub-confluent to confluent density, the medium was replaced with a CS-containing medium. Adherent bronchial cells were incubated with CS at 200 μg/cm$^2$ for 48 h. After incubation with CS, the medium was filtered through a 0.45 μm Millex filter unit (Advantech, Milpitas, CA, USA) to prepare CS CM. Similarly, BPDE CM was obtained from culture supernatants 48 h after incubation with BPDE at doses ranging from 500 to 1000 nM. The PMA-differentiated THP-1 macrophage cell line was incubated with or without 200 μg/cm$^2$ or 300 μg/cm$^2$ CS for 48 h, and culture supernatants were collected as THP-1 CS 48 h CM and THP-1 control 48 h CM, respectively, for use with 16HBE. The PMA-differentiated THP-1 macrophage cell line was incubated with or without 200 or 300 μg/cm$^2$ CS for 72 h, and culture supernatants were collected as THP-1 CS 72 h CM and THP-1 control 72 h CM, respectively, for use with NL20 and BEAS-2B. The VA13 fibroblast cell line was incubated with or without 200 μg/cm$^2$ CS for 48 h, and the culture supernatants were collected as VA13 CS CM and VA13 control CM. PMA-differentiated THP-1 cells were incubated with or without BPDE at doses ranging from 500 to 1000 nM for 48 h, and the culture supernatants were collected as THP-1 BPDE CM and THP-1 control CM. CM was either used immediately or stored at −80°C until further use.

## Cell proliferation assays for anchorage-independent growth

Bronchial cells were seeded in CS CM and BPDE CM with their corresponding control CM and incubated in poly-(HEMA)-coated 96-well plates in quadruplicate. After incubation,

growth was quantified using the 3-(4,5-dimethylthiazol-2-yl)-2,5-diphenyltetrazolium bromide (MTT) assay, according to the manufacturer's instructions (Promega, Madison, WI, USA). Briefly, cell suspensions in 96-well plates were stained with 15 μL of MTT solution for 4 h, allowing viable cells to convert MTT into the insoluble formazan product. The formazan products were dissolved in 100 μL of the solubilization/stop solution at 37˚C for 1 h. The absorbance of the solubilized mixture was then measured at 562 nm with a reference absorbance of 690 nm on a 96-well plate reader. The results indicative of anchorage-independent growth were calculated in at least three independent experiments.

## Quantification of soluble factors using ELISA

Quantikine ELISA human EGF, TNF-α, TGF-β1, and IL-6 immunoassay kits were purchased from R&D Systems (R&D Systems Inc., Minneapolis, MN, USA), and used to measure the concentration of soluble factors released from cells into culture supernatants. Bronchial cell lines were inoculated in the CM in a poly-(HEMA)-coated 6-well plate. After 4–6 days of incubation, the culture supernatants were collected and the secretion of EGF, TNF-α, TGF-β1, and IL-6 was measured in duplicate using ELISA, according to the manufacturer's instructions. The concentration of each soluble factor was calculated from the absorbance of the samples at 450 nm using a microplate reader.

## Western blot analysis

Cells were collected and lysed in radioimmunoprecipitation assay (RIPA) buffer. The extracted whole-cell protein lysates (15 μg each) were subjected to western blotting. The lysates were separated using SDS-PAGE with a 4%–12% gradient gel, followed by electroblotting of the separated proteins onto polyvinylidene difluoride (PVDF) membranes (GE Healthcare, Little Chalfont, UK). The blots were incubated in blocking buffer (Tris-buffered saline with 5% non-fat milk) at 4˚C overnight with the following primary antibodies: anti-cyclin A2 (H-432, Santa Cruz Biotechnology), anti-cdc2 (17, Santa Cruz Biotechnology), anti-c-Myc (E5Q6W, Cell Signaling Technology), anti-BRD4 (E2A7X, Cell Signaling Technology), anti-EZH2 (162666, BD Transduction Laboratories), anti-PARP1 (46D11, Cell Signaling Technology), anti-γH2AX (20E3, Cell Signaling Technology), and anti-actin (A-2066, Sigma-Aldrich). Immunoreactive proteins were detected by enhanced chemiluminescence after incubation with a horseradish peroxidase-conjugated secondary antibody.

## Statistical analysis

Differences were evaluated using *F*-test for equality of variance, followed by Student's *t*-test or Welch's *t*-test, where appropriate. The statistical calculations were performed using Microsoft Excel 2016 (Microsoft Corporation, Redmond, WA, USA). Statistical significance was set at $p < 0.05$. Data are presented as mean ± standard deviation (SD).

# Results

## Autocrine and paracrine CS CM promoted anchorage-independent growth of bronchial cell lines

We tested whether NL20, BEAS-2B, and 16HBE cells presented enhanced anchorage-independent growth in CS CM. Bronchial cells seeded in poly-(HEMA)-coated plates were incubated with CS CM or control CM for three to six days to assess anchorage-independent growth. All bronchial cell lines presented significantly accelerated anchorage-independent growth in

autocrine CS CM. All bronchial cell lines presented significantly enhanced anchorage-independent growth in THP-1 CS 72 h CM and THP-1 CS 48 h CM, but not in VA13 CS CM (Fig 1).

## Autocrine and paracrine BPDE CM promoted anchorage-independent growth of bronchial cell lines

All bronchial cell lines showed significantly accelerated anchorage-independent growth in autocrine BPDE CM and paracrine THP-1 BPDE CM (Fig 1).

## Suppression of bronchial cell line growth after exposure to CS showed enhanced anchorage-independent growth with CS and BPDE CM in autocrine and paracrine manners

After direct exposure to CS, bronchial cells have been reported to undergo apoptosis and DNA double-strand breaks [21, 22]. We tested whether bronchial cell lines after direct exposure to CS could promote anchorage-independent growth in CS and BPDE CM. Adherent NL20 and BEAS-2B cells were exposed to CS at doses of 200 $\mu g/cm^2$ and 50 $\mu g/cm^2$ for 12 h, and adherent 16HBE cells were exposed to 200 $\mu g/cm^2$ CS for 24 h. The growth of adherent bronchial cell lines was significantly suppressed after direct exposure to CS (Fig 2A). The bronchial cells exposed to CS for 12 or 24 h were immediately seeded in poly-(HEMA)-coated plates with CS or BPDE CM. The nonadherent bronchial cell lines pre-exposed to CS were subjected to MTT assay. Five days after incubation with CS and BPDE CM, all bronchial cell lines compromised by direct exposure to CS exhibited significantly enhanced anchorage-independent growth compared with those in control CM (Fig 2B). Nonadherent NL20, BEAS-2B, and 16HBE exposed to CS also exhibited enhanced growth in VA13 CS CM (Fig 2B), as opposed to their intact counterparts in VA13 CS CM (Fig 1). CS-exposed bronchial cell lines also showed significantly enhanced anchorage-independent growth in THP-1 CS 72 h CM, THP-1 CS 48 h CM, and THP-1 BPDE CM (Fig 2B).

## Increased expression of cyclin A2 and cdc2 in CS-exposed nonadherent bronchial cell lines after incubation with CS and BPDE CM

The expression of cell cycle-related proteins was studied in nonadherent bronchial cell lines incubated with CS or BPDE CM. The expression of cyclin A2 and cdc2 was comparable or slightly increased in intact nonadherent NL20, BEAS-2B, and 16HBE cells after incubation with CS CM compared with that of the control CM. CS-exposed nonadherent bronchial cell lines treated with CS CM showed higher expression of cyclin A2 and/or cdc2 than those treated with control CM (Fig 3). Both intact and CS-exposed nonadherent bronchial cell lines treated with BPDE CM exhibited upregulated expression of cyclin A2 and cdc2 compared with those treated with control CM (Fig 3).

## Elevated c-Myc expression in CS-exposed nonadherent bronchial cell lines treated with CS and BPDE CM

Overexpression of c-Myc has been implicated to play an important role in CS-, arsenite-, and hexavalent chromium-induced carcinogenesis [23–25]. We compared the expression of c-Myc in control and CS-exposed nonadherent bronchial cell lines treated with CS and BPDE CM. The expression of c-Myc was induced in CS-exposed nonadherent bronchial cell lines after incubation with CS and BPDE CM, except for CS-exposed nonadherent 16HBE treated with BPDE CM (Fig 4A). The expression of c-Myc, an early response gene to external stressors, was

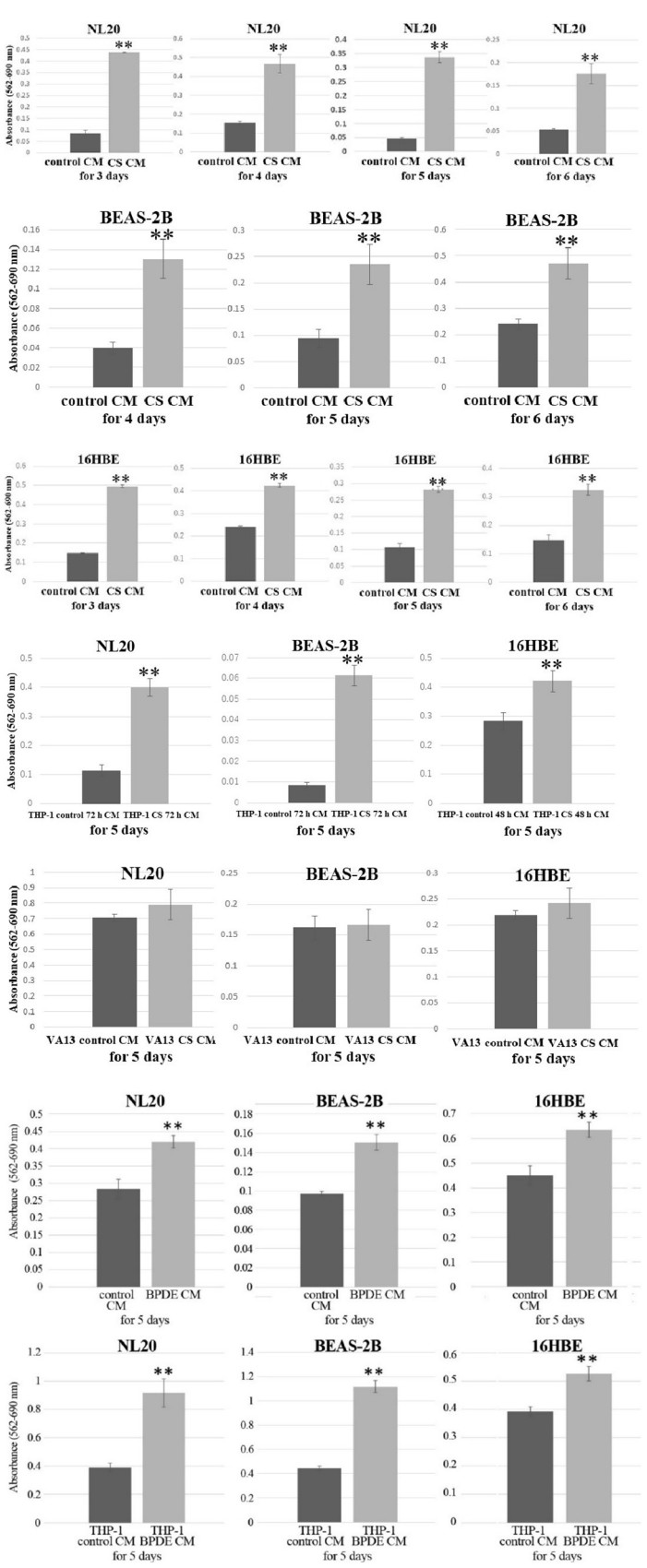

**Fig 1. Effects of CS and BPDE CM on anchorage-independent growth of bronchial cell lines.** MTT assay for nonadherent NL20, BEAS-2B, and 16HBE cells three to six days after incubation with autocrine CS CM, and five days after incubation with autocrine BPDE CM compared to the corresponding autocrine control CM. MTT assay for the nonadherent bronchial cell lines five days after incubation with paracrine CS CM (THP-1 CS 72 h CM, THP-1 CS 48 h CM, and VA13 CS CM) and paracrine BPDE CM (THP-1 BPDE CM) compared to the corresponding paracrine control CM. Asterisks indicate a significant difference between nonadherent bronchial cell lines cultured in CS and BPDE CM and those cultured in the corresponding control CM (*p < 0.05, **p < 0.01). Relative values are shown as mean ± SD from at least three independent experiments.

examined in adherent bronchial cell lines after exposure to CS and BPDE for 24 or 48 h. c-Myc expression was upregulated in adherent NL20, BEAS-2B, and 16HBE cells after direct exposure to CS. (Fig 4B).

## Enhanced expression of BRD4 and EZH2 in CS-exposed nonadherent bronchial cell lines treated with autocrine CS and BPDE CM

As transcriptional and epigenetic regulators and oncogenes *BRD4* and *EZH2* have been reported to be upstream activators of *c-Myc* and downstream *c-Myc* target genes, respectively [26], we investigated the expression of BRD4 and EZH2 in intact and CS-exposed nonadherent bronchial cell lines after incubation with CS and BPDE CM. While the expression of BRD4 was slightly elevated in nonadherent 16HBE treated with CS CM, CS-exposed nonadherent NL20, BEAS-2B, and 16HBE treated with CS CM exhibited enhanced expression of BRD4. Similarly, while the expression of BRD4 was unaltered or slightly increased in intact nonadherent bronchial cell lines treated with BPDE CM, all CS-exposed nonadherent bronchial cell lines treated with BPDE CM manifested elevated BRD4 expression. Although EZH2 expression was unaltered in intact nonadherent bronchial cell lines treated with CS CM, EZH2 was upregulated in CS-exposed nonadherent bronchial cell lines treated with CS CM. EZH2 expression was consistently increased in intact and CS-exposed nonadherent bronchial cell lines treated with BPDE CM (Fig 5A). BRD4, EZH2, cyclin A2, and cdc2 levels were unchanged or downregulated in adherent bronchial cell lines after direct exposure to CS or BPDE (Fig 5B).

Protein expression was evaluated in intact and CS-exposed nonadherent bronchial cell lines using the corresponding paracrine CM. Upregulated proteins varied among intact and CS-exposed nonadherent bronchial cell lines treated with THP-1 CS 72 h CM, THP-1 CS 48 h CM, and VA13 CS CM, apparently dependent on each bronchial cell line. However, CS-exposed nonadherent bronchial cell lines treated with THP-1 BPDE CM showed elevated levels of BRD4, EZH2, cyclin A2, and cdc2 more broadly than those treated with THP-1 CS 72 h CM, THP-1 CS 48 h CM, and VA13 CS CM (Fig 5C).

## Increased PARP1 expression in CS-exposed nonadherent bronchial cell lines treated with autocrine CS and BPDE CM

As CS and BPDE CM were found to elevate the anchorage-independent growth of bronchial cell lines even after growth suppression upon direct exposure to CS, we investigated the expression of PARP1, which has been documented to facilitate the repair of DNA damage, including DNA double-strand breaks, and/or mediate inflammation-driven carcinogenesis [27, 28]. We observed increased PARP1 expression in CS-exposed nonadherent bronchial cell lines after incubation with autocrine CS and BPDE CM (Fig 6). Among the proteins with increased expression in CS-exposed nonadherent bronchial cell lines treated with CS and BPDE CM, BRD4 has been reported to promote the repair of DNA double-strand breaks [29].

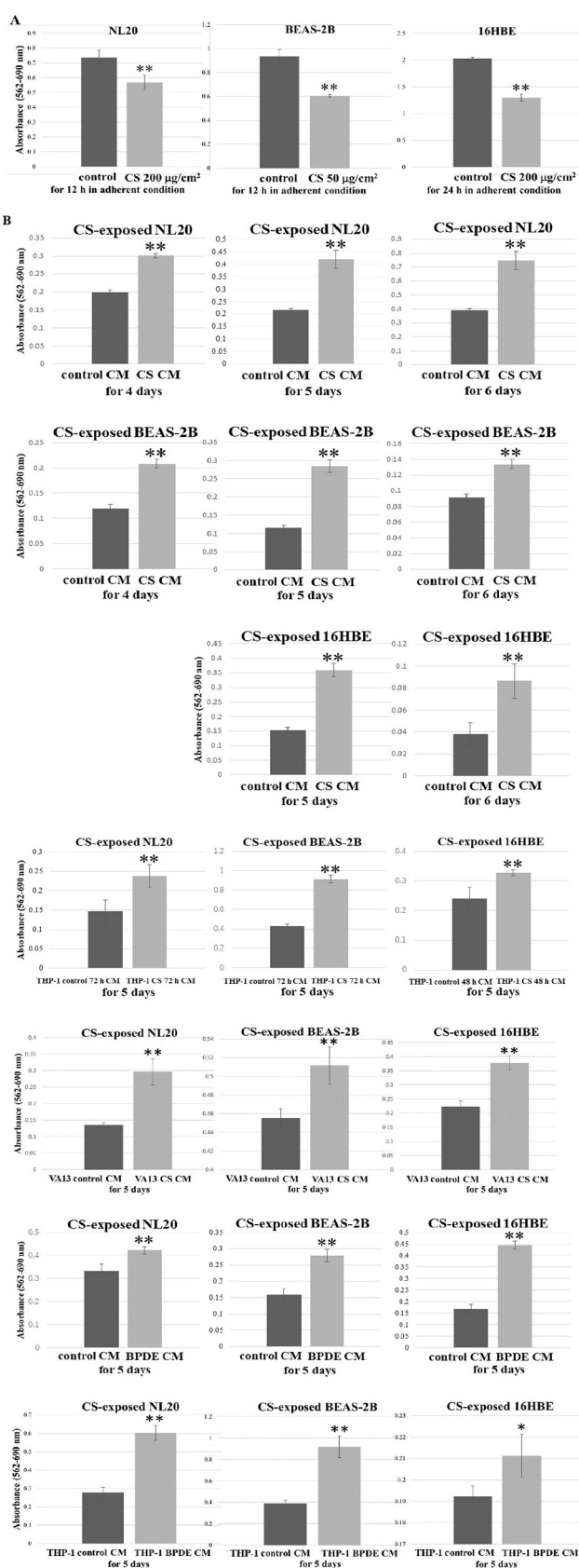

**Fig 2. CS and BPDE CM enhanced anchorage-independent growth of bronchial cell lines pre-exposed to CS.** (A) Adherent NL20 and BEAS-2B were exposed to CS at 200 μg/cm$^2$ and 50 μg/cm$^2$, and adherent 16HBE was exposed to CS at 200 μg/cm$^2$. Adherent NL20 and BEAS-2B were subjected to MTT assay after being exposed to CS for 12 h. Adherent 16HBE cells were subjected to MTT assay after being exposed to CS for 24 h. Asterisks indicate a significant difference between adherent bronchial cell lines after exposure to CS and those unexposed to CS (**p < 0.01). (B) CS-exposed and growth-suppressed nonadherent NL20, BEAS-2B, and 16HBE were subjected to MTT assay after four to six days of incubation with CS and BPDE CM and their corresponding control CM. Asterisks indicate a significant difference between CS-exposed nonadherent bronchial cell lines in CS and BPDE CM and those in the control CM (*p < 0.05, **p < 0.01). Relative values are shown as mean ± SD from three independent experiments.

We confirmed that the overexpression of BRD4 was concurrent with the upregulation of PARP1 in CS-exposed nonadherent bronchial cell lines treated with CS and BPDE CM. Therefore, we investigated the expression of γHA2X, a surrogate marker of DNA double-strand breaks. The expression of γHA2X was diverse and not necessarily inversely correlated with the expression levels of PARP1 and BRD4 in CS-exposed nonadherent bronchial cell lines treated with CS and BPDE CM. An increased expression of γHA2X was occasionally observed despite

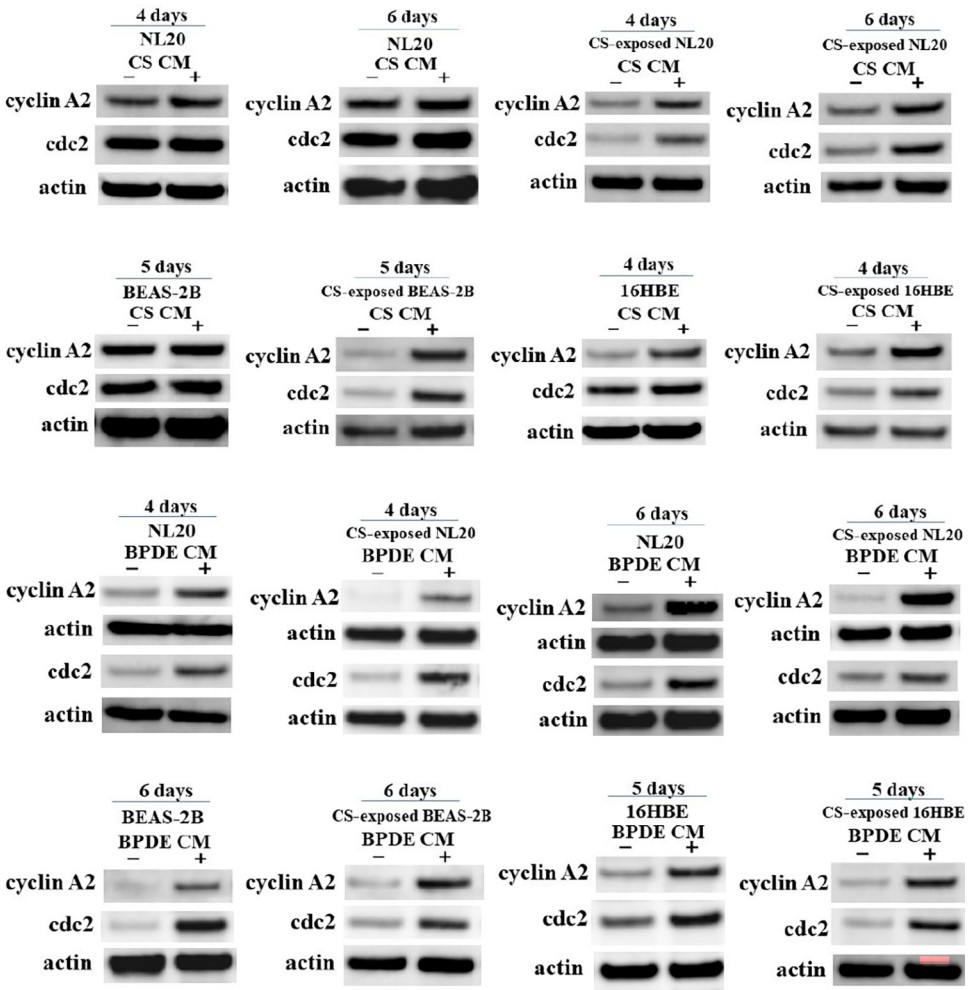

**Fig 3. Effects of autocrine CS and BPDE CM on the expression of cyclin A2 and cdc2 in intact and CS-exposed nonadherent bronchial cell lines.** Expression of cell cycle-related proteins cyclin A2 and cdc2 were studied using western blotting for intact and CS-exposed nonadherent bronchial cell lines after incubation with autocrine CS CM, BPDE CM, and their corresponding control CM.

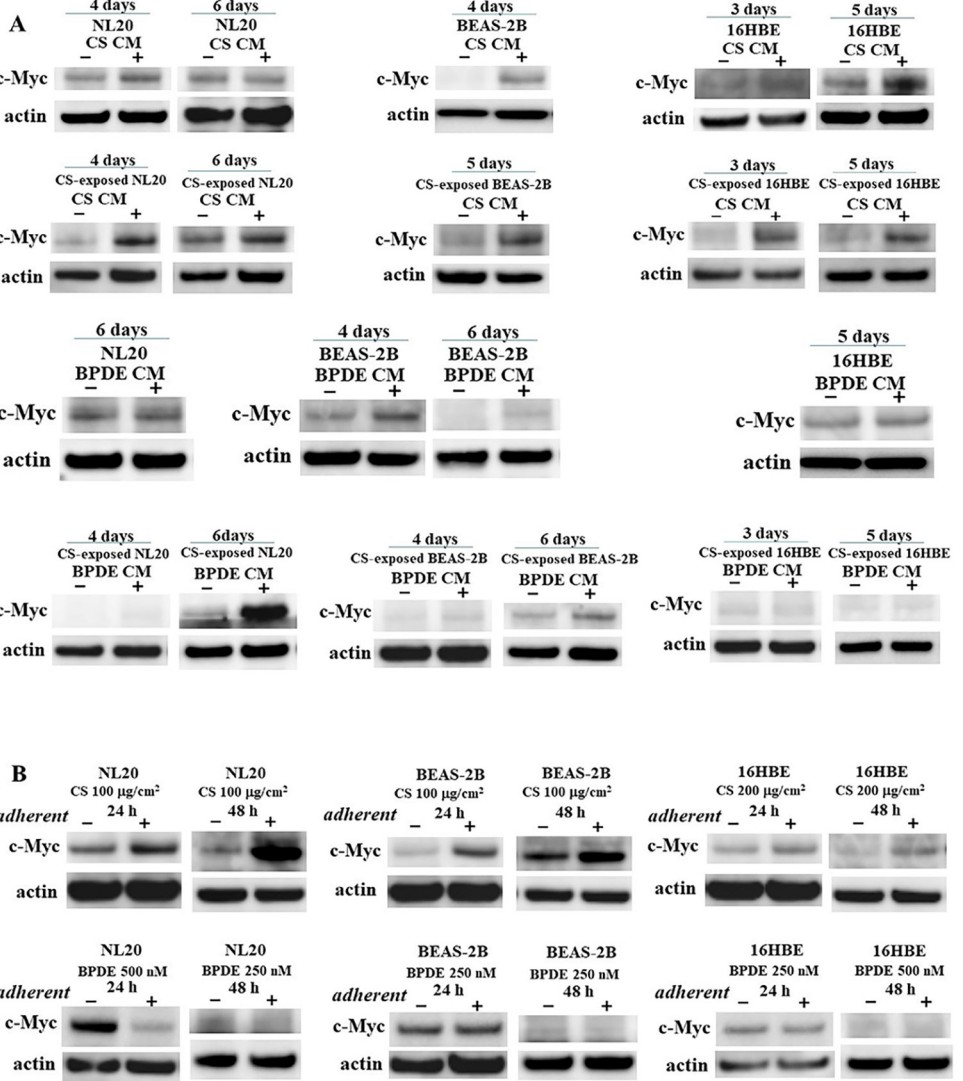

**Fig 4. Expression of c-Myc in CS CM- and BPDE CM-incubated nonadherent bronchial cell lines and the corresponding adherent bronchial cell lines after exposure to CS and BPDE.** (A) Expression of c-Myc in intact and CS-exposed nonadherent bronchial cell lines after incubation with autocrine CS CM, BPDE CM, and their corresponding autocrine control CM. (B) Expression of c-Myc in adherent bronchial cell lines 24 or 48 h after exposure to CS and BPDE.

overexpression of PARP1 and BRD4 in CS-exposed nonadherent bronchial cell lines treated with CS or BPDE CM (Fig 6), indicating that CS and BPDE CM enable CS-exposed nonadherent bronchial cell lines even with increased DNA double-strand breaks to accelerate proliferation.

## Increased concentration of soluble factors in culture supernatants of CS CM-incubated nonadherent bronchial cell lines

As CS and BPDE CM promoted the growth of nonadherent bronchial cell lines along with upregulated proteins, we tried to identify soluble factors in CM that contribute to the acceleration of growth and upregulated proteins. According to previous studies on CS-induced upregulation of soluble factors, we measured the concentrations of EGF, TNF-α, IL-6, and TGF-β1

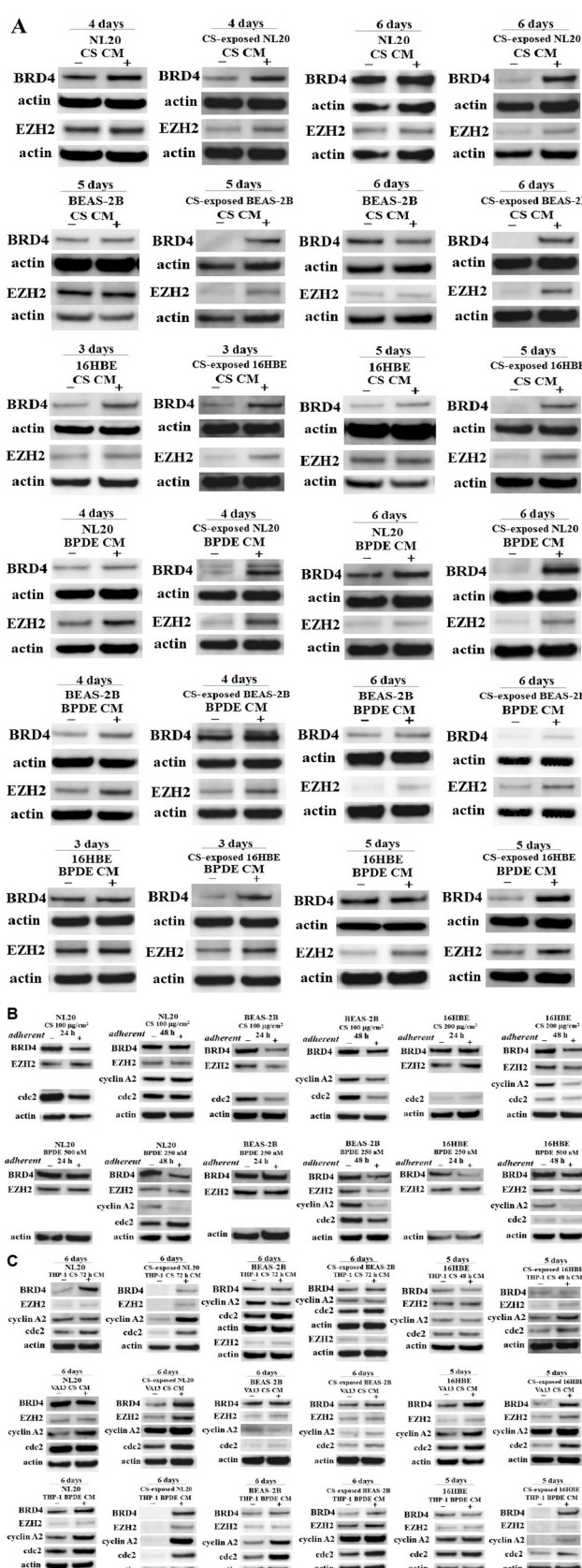

**Fig 5. Expression of BRD4, EZH2, cyclin A2, and cdc2 in CS CM- and BPDE CM-incubated intact and CS-exposed nonadherent bronchial cell lines, and the corresponding adherent bronchial cell lines after exposure to CS and BPDE.** (A) Expression of BRD4 and EZH2 in intact and CS-exposed nonadherent bronchial cell lines after incubation with autocrine CS and BPDE CM, compared with that in autocrine control CM. (B) Expression of BRD4, EZH2, cyclin A2, and cdc2 24 or 48 h after direct exposure to CS or BPDE in adherent bronchial cell lines. (C) Expression of BRD4, EZH2, cyclin A2, and cdc2 in intact and CS-exposed nonadherent bronchial cell lines after incubation with paracrine CS CM (THP-1 CS 72 h CM, THP-1 CS 48 h CM, and VA13 CS CM) and paracrine BPDE CM (THP-1 BPDE CM), compared with that in the corresponding paracrine control CM.

in culture supernatants. EGF concentration increased in the culture supernatants of CS CM- and BPDE CM-incubated nonadherent NL20 and BEAS-2B cells. TNF-$\alpha$ increased in the culture supernatants of nonadherent 16HBE cells after incubation with CS and BPDE CM. IL-6 was upregulated in the culture supernatants of all CS CM- and BPDE CM-incubated nonadherent bronchial cell lines. The EGF and TNF-$\alpha$ levels were also higher in the culture supernatants of adherent bronchial cell lines 48 h after exposure to CS. Similarly, the IL-6 level increased after exposure to CS for 48 h. The TGF-$\beta$1 level was not elevated in the culture supernatants of nonadherent bronchial cell lines treated with CS CM (Fig 7A). The results suggest that EGF in NL20 and BEAS-2B cells, TNF-$\alpha$ in 16HBE cells, and IL-6 in all bronchial cell lines were upregulated during incubation with the respective CS CM.

## Administration of recombinant human EGF, TNF-$\alpha$, and IL-6 promoted anchorage-independent growth of the corresponding bronchial cell lines

To explore the relationship between the soluble factors upregulated in culture supernatants and the growth of bronchial cell lines in CS and BPDE CM, recombinant human EGF, TNF-$\alpha$, or IL-6 was applied to the corresponding bronchial cell lines under nonadherent conditions. The administration of recombinant EGF promoted anchorage-independent growth of NL20 and BEAS-2B cells. Recombinant TNF-$\alpha$ caused 16HBE cells to increase anchorage-independent growth. Recombinant IL-6 also stimulated the nonadherent growth of bronchial cell lines (Fig 7B).

We tested whether NtAb against EGF, TNF-$\alpha$, and IL-6 counteracted the CS CM-induced anchorage-independent growth of bronchial cell lines. NtAbs against EGF and TNF-$\alpha$ significantly suppressed anchorage-independent growth of CS CM-incubated bronchial cell lines

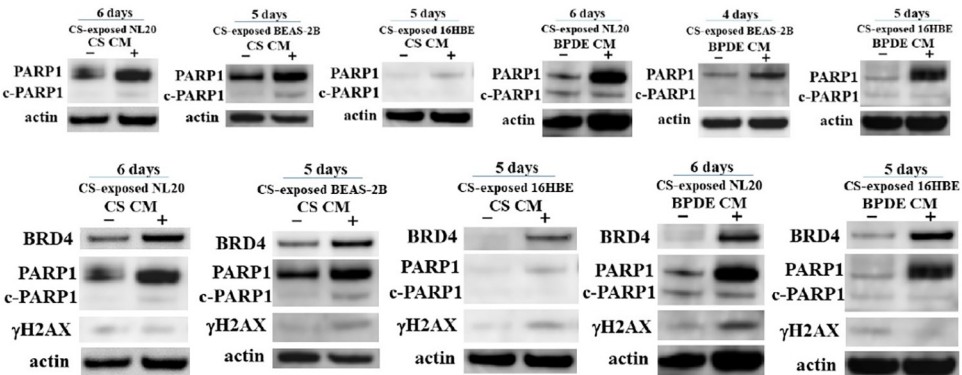

**Fig 6. Expression of PARP1, BRD4, and $\gamma$H2AX in autocrine CS CM- and BPDE CM-incubated CS-exposed nonadherent bronchial cell lines.** Expression of PARP1 in CS-exposed nonadherent bronchial cell lines after incubation with autocrine CS and BPDE CM compared with that in cells after incubation with the autocrine control CM. Expression of $\gamma$H2AX compared with that of BRD4 and PARP1 in CS-exposed nonadherent bronchial cell lines treated with autocrine CS CM, BPDE CM, and autocrine control CM. c-PARP1; cleaved PARP1.

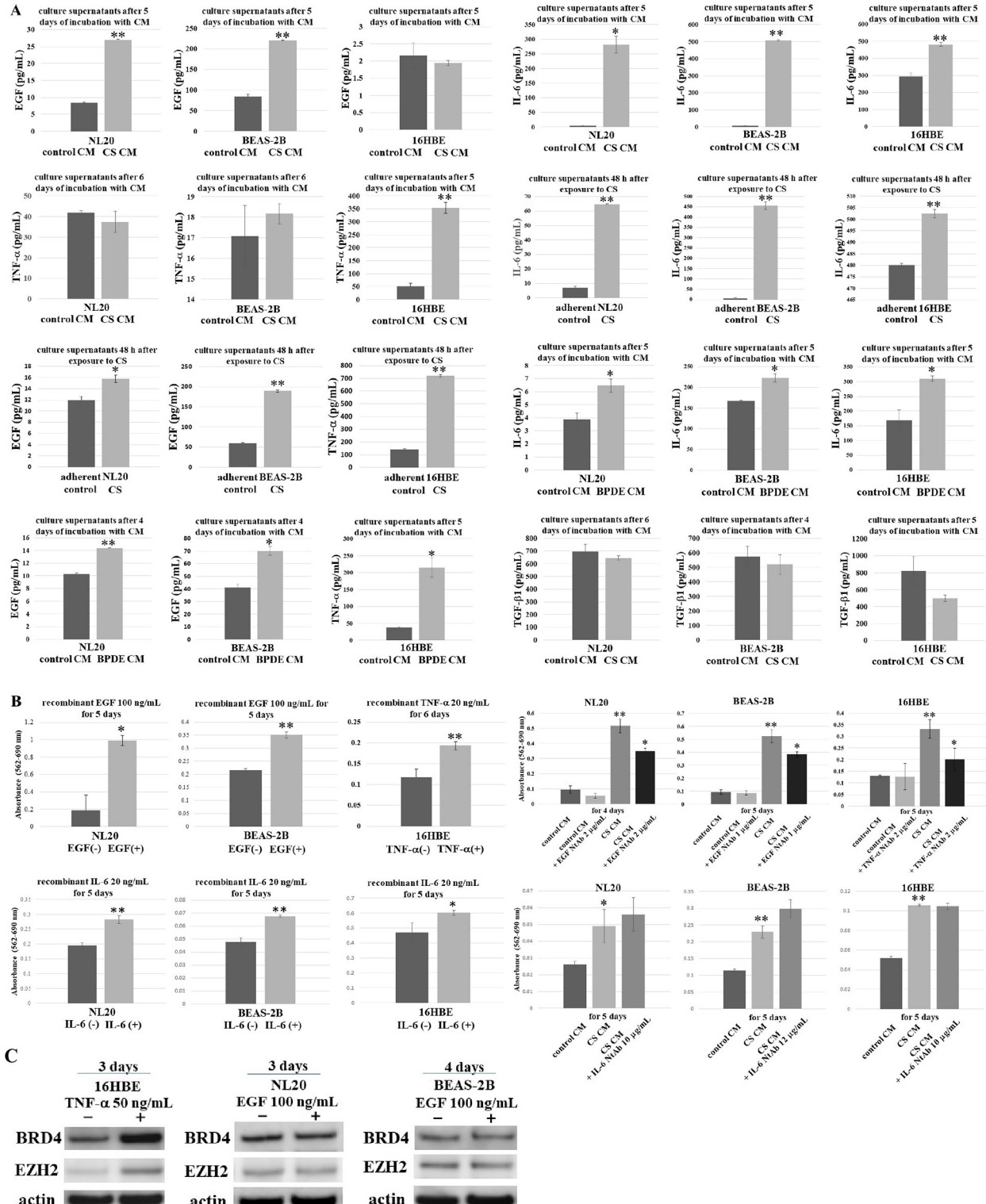

**Fig 7. Contribution of increased soluble mediators in culture supernatants to anchorage-independent growth of the corresponding CS CM-incubated bronchial cell lines.** (A) ELISA of EGF, TNF-α, IL-6, and TGF-β1 was performed with the culture supernatants of nonadherent NL20, BEAS-2B, and 16HBE 4–6 days after incubation with autocrine CS or BPDE CM compared with the autocrine control CM. ELISA of EGF, TNF-α, and IL-6 was also conducted with the culture supernatants 48 h after exposure to CS in adherent bronchial cell lines. Asterisks indicate a significant difference in concentration (*p < 0.05, **p < 0.01). Relative values are shown as mean ± SD from three independent experiments. (B) Anchorage-

independent growth of NL20, BEAS-2B, and 16HBE cells was measured using MTT 5–6 days after the administration of recombinant human EGF, TNF-α, and IL-6. The MTT assay was performed for autocrine CS CM-incubated nonadherent NL20 and BEAS-2B cells four and five days after EGF-neutralizing antibody (NtAb) treatment, respectively, and for autocrine CS CM-incubated nonadherent 16HBE cells, five days after TNF-α NtAb treatment. The MTT assay was also performed for autocrine CS CM-incubated nonadherent bronchial cell lines five days after IL-6 NtAb administration. Asterisks indicate a significant difference in growth (*p < 0.05, **p < 0.01). Relative values are shown as mean ± SD from three independent experiments. (C) Expression of BRD4 and EZH2 in nonadherent 16HBE cells three days after the administration of recombinant human TNF-α determined using western blotting and the expression of BRD4 and EZH2 3–4 days after treatment of nonadherent NL20 and BEAS-2B cells with recombinant human EGF.

(Fig 7B). EGF and TNF-α may contribute to the CS CM-induced increase in anchorage-independent growth of bronchial cell lines. Furthermore, the expression of BRD4 and EZH2 was induced after the treatment of nonadherent 16HBE cells with recombinant human TNF-α (Fig 7C), suggesting that TNF-α may be involved in CS CM- and BPDE CM-mediated upregulation of BRD4 and EZH2 in nonadherent 16HBE cells.

## Discussion

In this study, CS-exposed and growth-suppressed bronchial epithelial cells presented significantly enhanced anchorage-independent growth after incubation with autocrine and paracrine CS CM, and autocrine and paracrine BPDE CM. This result may resolve the discrepancy in counteractive cellular responses between CS-induced toxicity that affects growth and CS-induced inflammatory microenvironment that promotes growth. CS-exposed nonadherent bronchial cell lines treated with CS CM exhibited a more pronounced and extensive overexpression of cyclin A2 and cdc2 than their intact counterparts treated with CS CM. Furthermore, c-Myc expression increased in adherent bronchial cells after direct exposure to CS, implicating that c-Myc is an early response gene. However, c-Myc was mostly upregulated in CS-exposed nonadherent bronchial cells treated with CS CM and BPDE CM. Therefore, the upregulation of c-Myc in CS-exposed nonadherent bronchial cells with CS and BPDE CM should be considered separately from that of c-Myc upon direct exposure to CS. The *c-Myc* gene is a target gene of *BRD4* and an upstream effector gene of *EZH2* [22], and it is involved in the CS-induced transformation of rodent cell lines [19]. Therefore, further studies are required to elucidate its contribution to CS-induced carcinogenesis.

CS-exposed nonadherent bronchial cell lines showed upregulated and extensive expression of BRD4 and EZH2 when treated with CS CM compared with their intact counterparts treated with CS CM (Table 1). While CS-exposed nonadherent bronchial cell lines with BPDE CM showed an increase in BRD4 expression more markedly and extensively than intact cells with BPDE CM, a consistent increase in the expression of EZH2, cyclin A2, and cdc2 was observed in both intact and CS-exposed nonadherent bronchial cell lines treated with BPDE CM (Table 1).

In contrast to our *in vitro* study using autocrine CS CM, we did not necessarily obtain consistent results using paracrine CS CM with a few of the variations depending on bronchial epithelial cell lines, including intact *versus* CS-exposed cells and/or paracrine cell lineages used for CM. As paracrine factors are more complex and difficult to evaluate (e.g., dendritic cells, T lymphocytes, and other immunocompetent cells and their interactions) than autocrine factors, experiments using 3D cell culture or *in vivo* studies are warranted. However, bronchial cell lines exposed to CS in paracrine CS and BPDE CM exhibited protein overexpression that was more extensive or more pronounced than their intact counterparts in paracrine CS and BPDE CM.

No obvious differences were observed in the degree of nonadherent growth stimulation between intact bronchial cell lines and those exposed to CS with CS and BPDE CM, despite the extensive overexpression of the analyzed proteins in CS-exposed nonadherent bronchial

**Table 1. (1) List of protein expression patterns in intact or CS-exposed nonadherent bronchial cell lines after incubation with CS or BPDE CM.** (2) List of protein expression patterns in adherent bronchial cell lines after direct exposure to CS or BPDE.

| Condition of cell line | Treatment with autocrine CM | (1) Protein expression patterns | | | | |
|---|---|---|---|---|---|---|
| | | cyclin A2 | cdc2 | BRD4 | EZH2 | c-Myc |
| Nonadherent NL20 | CS CM | unchanged expression | unchanged expression | unchanged expression | unchanged expression | slightly increased expression |
| | BPDE CM | increased expression | increased expression | unchanged expression | slightly increased expression | unchanged expression |
| CS-exposed nonadherent NL20 | CS CM | increased expression | increased expression | markedly increased expression | increased expression | increased expression |
| | BPDE CM | markedly increased expression | increased expression | markedly increased expression | markedly increased expression | increased expression |
| Nonadherent BEAS-2B | CS CM | unchanged expression | unchanged expression | unchanged expression | unchanged expression | increased expression |
| | BPDE CM | increased expression | markedly increased expression | slightly increased expression | increased expression | increased expression |
| CS-exposed nonadherent BEAS-2B | CS CM | markedly increased expression | markedly increased expression | markedly increased expression | markedly increased expression | increased expression |
| | BPDE CM | markedly increased expression | markedly increased expression | increased expression | increased expression | increased expression |
| Nonadherent 16HBE | CS CM | increased expression | unchanged expression | increased expression | unchanged expression | slightly increased expression |
| | BPDE CM | increased expression | slightly increased expression | unchanged expression | increased expression | unchanged expression |
| CS-exposed nonadherent 16HBE | CS CM | increased expression | slightly increased expression | markedly increased expression | markedly increased expression | increased expression |
| | BPDE CM | markedly increased expression | markedly increased expression | markedly increased expression | increased expression | unchanged expression |
| Condition of cell line | Direct exposure | (2) Protein expression patterns | | | | |
| | | cyclin A2 | cdc2 | BRD4 | EZH2 | c-Myc |
| Adherent NL20 | CS | | decreased expression | decreased expression | unchanged expression | increased expression |
| | BPDE | decreased expression | unchanged expression | decreased expression | decreased expression | decreased expression |
| Adherent BEAS-2B | CS | decreased expression | decreased expression | decreased expression | decreased expression | increased expression |
| | BPDE | decreased expression | decreased expression | decreased expression | decreased expression | unchanged expression |
| Adherent 16HBE | CS | decreased expression | decreased expression | decreased expression | decreased expression | increased expression |
| | BPDE | decreased expression | unchanged expression | decreased expression | decreased expression | unchanged expression |

cell lines treated with CM. The underlying reasons could be as follows. First, CS-exposed nonadherent bronchial cell lines treated with CS and BPDE CM may require higher expression of growth-driving proteins than intact ones to overcome CS-induced cytotoxicity. Second, CS-exposed nonadherent bronchial cell lines with CS and BPDE CM may require extensive over-expression of BRD4 and PARP1 to repair CS-induced DNA double-strand breaks by error-prone non-homologous end-joining (NHEJ) pathways [27, 28] to maintain accelerated growth. We demonstrated that γH2AX, a marker of DNA double-strand breaks, was occasionally upregulated despite the overexpression of BRD4 and PARP1 in CS-exposed nonadherent bronchial cell lines treated with CS or BPDE CM. This result suggests the presence of overgrown bronchial cells with DNA double-strand breaks in CS-exposed nonadherent bronchial cell lines treated with CS or BPDE CM. Simultaneously, γH2AX was not always upregulated in

correlation with the overexpression of BRD4 and PARP1 in CS-exposed nonadherent bronchial cell lines treated with CS or BPDE CM. Therefore, a single CS or BPDE CM treatment is sufficient to induce the growth of CS-exposed nonadherent bronchial cells but may not be sufficient to induce the genetic damage in CS-exposed overgrown nonadherent bronchial cells. Further investigation is warranted to demonstrate whether NHEJ-related genes besides γH2AX are aberrantly expressed in bronchial cells undergoing both repeated exposure to CS and repeated treatment with CS and/or BPDE CM using western blot and fluorescent microscopy analyses. Ultimately, regarding the early events of mutagenesis and inflammation in CS-induced carcinogenesis, further studies are crucial to determine whether CS and BPDE CM accelerate CS-exposed nonadherent bronchial cell line proliferation while retaining genetic mutations caused by insufficient or aberrant DNA repair via error-prone NHEJ, despite the overexpression of BRD4 and PARP1.

We demonstrated that BPDE CM induced more extensive upregulation of the analyzed proteins than CS CM in CS-exposed nonadherent bronchial cell lines. However, this result with the use of BPDE CM was expected, as previous studies have reported that benzo(a)pyrene or its active metabolite BPDE induces inflammatory responses in human macrophage cells, fibroblast cells, and bronchial epithelial cells [30–32]. Moreover, BPDE-induced transformation reportedly mediates BPDE-induced inflammation with BPDE-induced mutagenesis [33]. Meanwhile, the upregulated soluble factors EGF, TNF-α, and IL-6 were consistent between the culture supernatants of nonadherent bronchial cell lines treated with CS CM and those treated with BPDE CM in our study. Therefore, further research is required to identify other mediators that contribute to the extensive increase in the expression of BRD4, EZH2, cyclin A2, and cdc2 in intact and CS-exposed nonadherent bronchial cell lines treated with BPDE CM. BPDE is considered more mutagenic than CS because it causes highly mutagenic bulky DNA adducts [34] besides DNA double-strand breaks [35]. Furthermore, PARP1 and BRD4 have been reported to facilitate inflammation and inflammation-driven carcinogenesis [36, 37]. Inflammation has also been documented to cause DNA damage, and vice versa [38–40]. Considering the above, BPDE-induced inflammation may exert an effect stronger than CS-induced inflammation, promoting CS-induced carcinogenesis. However, this study has not clearly indicated whether BPDE *per se* has a role in CS-induced carcinogenesis mediated by CS- or BPDE-induced inflammation.

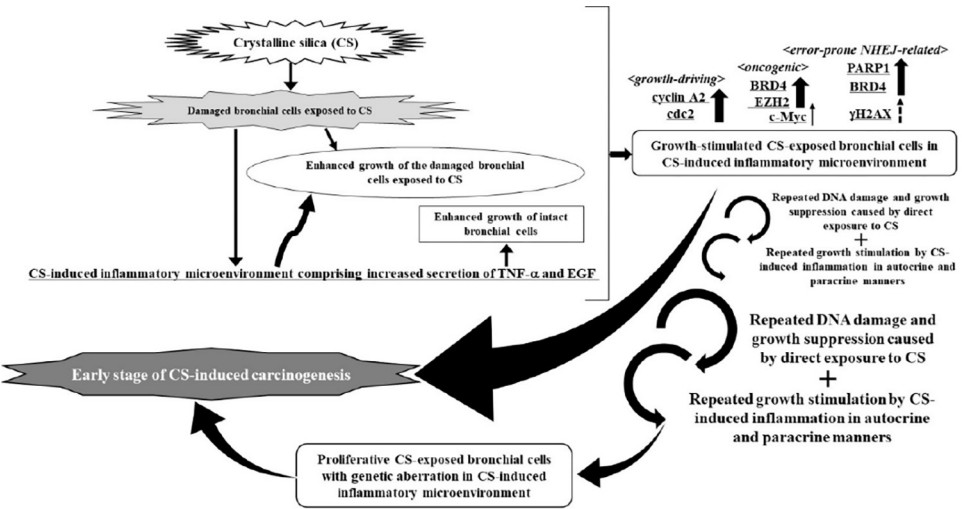

**Fig 8. Hypothetical pathway underlying CS-induced carcinogenesis through the cooperation of CS-induced genotoxicity with CS-induced inflammatory growth promotion of bronchial cells.**

## Conclusions

In this study, we demonstrated that CS and BPDE CM consisting of upregulated EGF or TNF-α enabled CS-damaged, and growth-suppressed nonadherent bronchial cell lines to not only proliferate but also provoke the expression of the oncogenic proteins BRD4, EZH2, PARP1, and c-Myc, despite occasional upregulation of γH2AX. This implies that CS-induced carcinogenesis may be aggravated by the cooperation of CS-induced inflammation and genotoxicity. This *in vitro* observation led us to infer that the combination of repetitive DNA damage caused by direct exposure to CS and repetitive growth stimulation by CS-induced inflammation could increase the probability of proliferative bronchial cells with genetic aberrations, eventually leading to lung cancer, as represented by a hypothetical pathway (Fig 8). However, a novel *in vitro* approach is required to reveal the transformation of bronchial epithelial cells through the cooperation of CS-induced genotoxicity with CS-induced inflammation.

## Supporting information

**S1 Raw images.**
(PDF)

## Acknowledgments

The authors would like to thank Prof. Dieter C. Gruenert for generously providing the human bronchial epithelial cell line, 16HBE14o-.

## Author Contributions

**Conceptualization:** Motoo Katabami.

**Data curation:** Motoo Katabami, Ichiro Kinoshita, Hirotoshi Dosaka-Akita.

**Formal analysis:** Motoo Katabami.

**Funding acquisition:** Ichiro Kinoshita, Hirotoshi Dosaka-Akita.

**Investigation:** Motoo Katabami.

**Methodology:** Motoo Katabami.

**Project administration:** Motoo Katabami, Ichiro Kinoshita, Shin Ariga, Yasushi Shimizu, Hirotoshi Dosaka-Akita.

**Resources:** Ichiro Kinoshita, Shin Ariga, Yasushi Shimizu, Hirotoshi Dosaka-Akita.

**Supervision:** Ichiro Kinoshita, Hirotoshi Dosaka-Akita.

**Writing – original draft:** Motoo Katabami.

**Writing – review & editing:** Motoo Katabami, Ichiro Kinoshita, Hirotoshi Dosaka-Akita.

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
