## [Decision Letter · Decision Letter 0]

30 Jan 2023

PONE-D-22-33801Crystalline silica-exposed human lung epithelial cells presented enhanced anchorage-independent growth with upregulated expression of BRD4 and EZH2 in autocrine and paracrine mannersPLOS ONE

Dear Dr. Kinoshita,

Thank you for submitting your manuscript to PLOS ONE. After careful consideration, we feel that it has merit but does not fully meet PLOS ONE’s publication criteria as it currently stands. Therefore, we invite you to submit a revised version of the manuscript that addresses the points raised during the review process.

We look forward to receiving your revised manuscript.

Kind regards,

Thierry Rabilloud

Academic Editor

PLOS ONE

Journal Requirements:

Reviewers' comments:

Reviewer's Responses to Questions

**Comments to the Author**

1. Is the manuscript technically sound, and do the data support the conclusions?

Reviewer #1: Partly

Reviewer #2: Yes

2. Has the statistical analysis been performed appropriately and rigorously? 

Reviewer #1: I Don't Know

Reviewer #2: Yes

3. Have the authors made all data underlying the findings in their manuscript fully available?

Reviewer #1: No

Reviewer #2: Yes

4. Is the manuscript presented in an intelligible fashion and written in standard English?

Reviewer #1: Yes

Reviewer #2: Yes

5. Review Comments to the Author

Reviewer #1: Interesting study, with a lot of data. The reading is dense, and the results highlight some interesting track of understanding of the crystalline silica effects.

Here are some points to be explained or modified for a better/easier comprehension:

Avoid abbreviation in the abstract (i.e. CM, BPDE, NtAbs), this would facilitate the reading.

Student's test is not so appropriate for a small number of replicate.

Materials and methods section:

references of the reagents and the cell line are missing,

Some information is not clear about the reagent (storage, concentration, sterilization...) the crystalline silica is sterilized under its solid form?

How the applied doses (CS, BPDE) have been chosen? Is it physiologically relevant compared with human smokers or workers/miners?

Results:

For the quantification of soluble factors, explain why the incubation time was 4-6 days.

The protocol of cell exposure for all the experiments needs to be clarified (5 days for anchorage-independent growth, 4-6 days for soluble factor quantification), all the incubation times make the comprehension difficult.

Is there a negative control for particle uptake? For example with Cyclin and cdc2 or c-myc, the expression is modified with the CS CM, is it due to CS or to the presence of particles inside the CM?

The gene expression was investigated, what about metabolism? Did the cells proliferate?

Discussion:

A scheme of cascade response should be added to facilitate the comprehension of the different actors (cMyc, cdc2, PARP1, gH2AX, BRD4, EZH2, EGF, TNFa, IL-6) Maybe the authors can add a summary table with the up/down-regulation for all of the target studied.

Figures:

Be sure that the size and the quality of the figures allow an easy reading

Fig 2: A) explain the 12h or 24h of exposure,

A and B) when it is possible, use similar absorbance scales (for comparison between cell types)

Fig 3

Same as previously, explain the 4-5-6 days of incubation

On the third line: modify legend - 4 days CS-exposed NL20 BPDE CM

Fig 5A and B: add the figure letters on the images.

Fig 6

Why is Fig6A duplicated in 6B? This is confusing - one figure (6B) will facilitate the reading

Fig 7B and 7C: add the figure letters

Fig 7C: TNF for 16HBE cells, EGF for NL20 and BEAS-2B cells, explain these choices

Reviewer #2: The authors presented clearly the subject of their research with various hypothesis which have been validated by relevant experiments.

Some minor corrections are provided below:

- There is no indication on the passages used for the different cell lines for the experiments performed and presented in this publications. This addition could provide additional insights in the understanding of the molecular mechanisms.

- The exposure is described as 5 days with some data at day 4 and day 6 but there is no indication of the typical cell doubling times of the different cell lines, and the potential imapct of the treatment. This addition could be useful to reinforce the conclusion presented in the article.

- in the article, the authors mentionned that "When the cells reached sub-confluent to confluent density, the medium was replaced with a CS-containing medium." Is there a strong impact for a same surface of the number of cells present to reach sub confluent to confluent density, and if yes is it a variable which should be adjusted to compare the data through the different cell models and conditions ?

- finally for gammaH2aX data, it could be useful to have some confocal images showing the potential variation in terms of foci density per nucleus ( or absence of this variation) to better assess the relevance of the presented data.

6. PLOS authors have the option to publish the peer review history of their article (what does this mean?). If published, this will include your full peer review and any attached files.

Reviewer #1: No

Reviewer #2: No

---

## [Author Response · Author response to Decision Letter 0]

29 Mar 2023

Response to Reviewers have been uploaded seperately as the response file contains figure and special characters which will not be displayed properly if pasted the response here.

---

## [Decision Letter · Decision Letter 1]

20 Apr 2023

Crystalline silica-exposed human lung epithelial cells presented enhanced anchorage-independent growth with upregulated expression of BRD4 and EZH2 in autocrine and paracrine manners

PONE-D-22-33801R1

Dear Dr. Kinoshita,

We’re pleased to inform you that your manuscript has been judged scientifically suitable for publication and will be formally accepted for publication once it meets all outstanding technical requirements.

Kind regards,

Thierry Rabilloud

Academic Editor

PLOS ONE

Additional Editor Comments (optional):

Reviewers' comments:

Reviewer's Responses to Questions

**Comments to the Author**

1. If the authors have adequately addressed your comments raised in a previous round of review and you feel that this manuscript is now acceptable for publication, you may indicate that here to bypass the “Comments to the Author” section, enter your conflict of interest statement in the “Confidential to Editor” section, and submit your "Accept" recommendation.

Reviewer #1: All comments have been addressed

Reviewer #2: All comments have been addressed

2. Is the manuscript technically sound, and do the data support the conclusions?

Reviewer #1: (No Response)

Reviewer #2: Yes

3. Has the statistical analysis been performed appropriately and rigorously? 

Reviewer #1: (No Response)

Reviewer #2: Yes

4. Have the authors made all data underlying the findings in their manuscript fully available?

Reviewer #1: (No Response)

Reviewer #2: Yes

5. Is the manuscript presented in an intelligible fashion and written in standard English?

Reviewer #1: (No Response)

Reviewer #2: Yes

6. Review Comments to the Author

Reviewer #1: (No Response)

Reviewer #2: The authors have modified their manuscript based on the first round of review and the improvement is significant. Its current format is ready for publication.

7. PLOS authors have the option to publish the peer review history of their article (what does this mean?). If published, this will include your full peer review and any attached files.

Reviewer #1: No

Reviewer #2: No

---

## [Editor Report · Acceptance letter]

28 Apr 2023

PONE-D-22-33801R1 

Crystalline silica-exposed human lung epithelial cells presented enhanced anchorage-independent growth with upregulated expression of BRD4 and EZH2 in autocrine and paracrine manners 

Dear Dr. Kinoshita:

I'm pleased to inform you that your manuscript has been deemed suitable for publication in PLOS ONE. Congratulations! Your manuscript is now with our production department. 

Kind regards, 

on behalf of

Dr. Thierry Rabilloud 

Academic Editor

PLOS ONE